# Oxidized Natural Biopolymer for Enhanced Surface, Physical and Mechanical Properties of Glass Ionomer Luting Cement

**DOI:** 10.3390/polym15122679

**Published:** 2023-06-14

**Authors:** Aftab Ahmed Khan, Ahmed Bari, Abdulaziz Abdullah Al-Kheraif, Hanan Alsunbul, Hind Alhaidry, Rasha Alharthi, Alhanoof Aldegheishem

**Affiliations:** 1Dental Biomaterials Research Chair, Dental Health Department, College of Applied Medical Sciences, King Saud University, Riyadh 11451, Saudi Arabia; aalkhuraif@ksu.edu.sa; 2Department of Pharmaceutical Chemistry, College of Pharmacy, King Saud University, Riyadh 11451, Saudi Arabia; abari@ksu.edu.sa; 3Restorative Dentistry Department, College of Dentistry, King Saud University, Riyadh 11451, Saudi Arabia; halumbol@ksu.edu.sa; 4Advanced General Dentistry, Prince Sultan Military Medical City, Riyadh 13514, Saudi Arabia; halhaidary@psmmc.med.sa; 5Clinical Dental Science Department, College of Dentistry, Princess Nourah Bint Abdulrahman University, Riyadh 11671, Saudi Arabia; rsalharthi@pnu.edu.sa (R.A.); asaldegheishem@pnu.edu.sa (A.A.)

**Keywords:** glass ionomer cement, Gum Arabic, luting cement, mechanical properties, physical properties

## Abstract

This laboratory investigation aimed to synthesize and characterize micron-sized Gum Arabic (GA) powder and incorporate it in commercially available GIC luting formulation for enhanced physical and mechanical properties of GIC composite. Oxidation of GA was performed and GA-reinforced GIC in 0.5, 1.0, 2.0, 4.0 & 8.0 wt.% formulations were prepared in disc-shaped using two commercially available GIC luting materials (Medicem and Ketac Cem Radiopaque). While the control groups of both materials were prepared as such. The effect of reinforcement was evaluated in terms of nano hardness, elastic modulus, diametral tensile strength (DTS), compressive strength (CS), water solubility and sorption. Two-way ANOVA and post hoc tests were used to analyze data for statistical significance (*p* < 0.05). FTIR spectrum confirmed the formation of acid groups in the backbone of polysaccharide chain of GA while XRD peaks confirmed that crystallinity of oxidized GA. The experimental group with 0.5 wt.% GA in GIC enhanced the nano hardness while 0.5 wt.% and 1.0 wt.% GA in GIC increased the elastic modulus compared to the control. The CS of 0.5 wt.% GA in GIC and DTS of 0.5 wt.% and 1.0 wt.% GA in GIC demonstrated elevation. In contrast, the water solubility and sorption of all the experimental groups increased compared to the control groups. The incorporation of lower weight ratios of oxidized GA powder in GIC formulation helps in enhancing the mechanical properties with a slight increase in water solubility and sorption parameters. The addition of micron-sized oxidized GA in GIC formulation is promising and needs further research for improved performance of GIC luting composition.

## 1. Introduction

Throughout the years, a wide variety of luting materials have been proposed and used in restorative dentistry. A range of provisional and long-term luting materials such as silicate cement, zinc oxide eugenol, zinc phosphate cement, resin-modified glass ionomer cement, polyalkenoate cement and resin composite are available [1]. However, among these materials available materials, polyalkenoate cement aka glass ionomer cement (GIC) is one such mainstream dental luting material.

In addition to adequate working and setting times, the desired attributes for any luting material for crown and bridge cementation include good bonding to tooth/restoration, biocompatibility, low solubility, anticariogenicity and appropriate marginal seal with high strength properties to resist torsional, shear and compressive forces during functioning [2]. Self-adhesive resin cement is a relatively new addition to luting materials, offering favourable properties [3]. However, it does not possess the same anti-cariogenic properties as conventional GIC or resin-modified GIC, and it is also intolerant to moisture [4]. 

In contrast, GIC has certain specific benefits over resin cement, including high biocompatibility with the pulp [5], chemical bonding with the tooth structure [6], fluoride release and recharging [7], which guard the tooth against the development of secondary caries. Yet, one of the most important drawbacks of GICs is their inherent lower mechanical properties [1]. Due to the inability to transfer stresses from crowns or fixed partial dentures to the tooth structure, clinical failure of the restoration is inevitable [8].

In recent years, there have been several endeavours to enhance the characteristics of GIC restorative materials through the incorporation of fibres into the mixture, aiming to enhance strength and increase the elasticity modulus. Nevertheless, there is a scarcity of studies concentrating on reinforcing the GIC luting material. Recently, an attempt was made to reinforce silver nanoparticles in GIC luting powder. However, the investigators observed non-significant results on microhardness and compressive strength [9]. According to a study conducted by Gupta et al. [10] and Saran et al. [11], incorporating a 10% of ceramic additive in GIC luting powder was found to increase the compressive strength. However, this improvement in properties came at the cost of longer setting time and increased film thickness.

Gum Arabic (GA), also known as Gum acacia, is a naturally obtained polymer with antimicrobial activity [12]. It is a non-toxic natural excipient commonly used for the sustained release of drugs due to its bioactive formula delivery capabilities [13]. GA is extracted from the hardened exudates of plants Acacia Senegal and Acacia Seyal [10]. It is biocompatible and considered an exceptional binder [13]. For clinical success, the luting material must exhibit mechanical stability during mastication and parafunctional habits [14,15]. Considering these traits, we presumed that the addition of micron-sized oxidized GA powder would provide improved bonding due to surface modification and introduction of functional groups. This may provide increased reactivity between GA and GIC particles and reduce the brittleness and prevent sudden crack propagation in GIC. 

The primary objective of this laboratory study was to modify the powder composition of GIC luting cement by incorporating oxidized GA powder in different weight percentages. The aim was to improve the mechanical characteristics of GIC luting. Specifically, the study focused on evaluating the compressive strength (CS) and diametral tensile strength (DTS) of the modified GIC luting through these commonly used tests in research. Additionally, the surface and physical properties of the experimental GIC luting were examined. The hypothesis was that the inclusion of GA powder in conventional GIC luting cement would enhance the aforementioned properties of the experimental GIC luting material.

## 2. Materials and Method

### 2.1. Oxidization of GA Powder

In 20 mL of distilled water, 1 g of GA powder was added and the mixture was heated at 70 °C and stirred for 30 min. Next, 30 mL of 30% H_2_O_2_ was added incrementally followed by a catalytic amount of FeSO_4_ (2 mg). The reaction mixture was heated at 100 °C for 2 h and during this time distilled water was added in increments to keep the overall volume of the mixture remains same. After completion of the reaction which was checked by peroxide strip test, the water was evaporated in a vacuum.

### 2.2. Fourier Transform Infrared (FTIR) Spectroscopy Analysis

All solvents and reagents were purchased from Aldrich Chemical Company. To check for the presence of peroxide in the mixture, peroxide strips were employed as a detection method. Furthermore, FTIR spectroscopy was conducted to analyze and identify the molecular components and structures of the GA powder following the oxidation process. The spectra were performed with the ATR attachment using a spectrometer (Alpha II, Bruker, Billerica, MA, USA). The GA powder was placed on the crystal of ATR-FTIR and the spectra were scanned in the range of 400 to 4000 cm^−1^.

### 2.3. X-ray Diffraction (XRD)

XRD evaluation of GA powder was performed for crystal size and phase detection using Ultima IV diffractometer (Rigaku Corporation, Tokyo, Japan) over 3–140° 2θ range at 2.0 deg./min scan speed. Based on the reported procedures, the crystallinity and crystallite were determined [16]. The tube anode was Cu with Ka = 0.154 nm monochromatized with a graphite crystal. The pattern was collected at tube voltage (40 kV) and tube current (40 mA) in step scan mode (step size 0.02°, counting time 1 s per step).

### 2.4. Preparation of Samples

Two commercially available type 1 GIC, i.e., Medicem (Promedica, Dental Material GmbH, Neumuenste, Germany) and Ketac Cem Radiopaque (3M ESPE, Seefeld/Oberbay, Germany) were obtained and used. To prepare experimental cement, oxidized GA powder of 50–150 microns in size was added in different weight % (0.5, 1.0, 2.0, 4.0, 8.0 wt.%) to the powder of each GIC used. The mixture was initially mixed manually and then placed on a vibrator for 5 min for optimal mixing. The control groups of each GIC brand were prepared with 0 wt.% of GA (Table 1). The samples of both the control and experimental GICs were mixed with the liquid according to the recommended powder:liquid ratio, i.e., 1:1.

The disk-shaped samples having dimensions of 6 mm diameter and 3 mm height were prepared from each study group (Figure 1). The samples were prepared by mixing the powder and liquid until a paste-like consistency was achieved and poured into a silicon mould. The samples were removed from the mould after 30 min and stored in labelled containers at 37 °C in an incubator for 24 h and subjected to tests to evaluate the physical and mechanical properties. All samples were prepared by a single trained operator at room temperature (23 °C).

### 2.5. Nanoindentation Test

A nanomechanical tester (UMT1, Bruker, Santa Barbara, CA, USA) with a Berkovich diamond indenter tip having a nominal radius of 100 nm was used to make the nanoindentations. The system was calibrated to produce an accurate indenter area function and correct instrument compliance. The experiments on study samples (*n* = 8) were conducted at a room temperature of 23 °C, with loading and unloading rates of 2.0 mN/s and a dwell time of 10 s during the peak load period. The maximal load was set to 20.0 mN. Each sample yielded a total of 3 measurements, and the mean value of nano hardness and elastic modulus for each sample was computed [17].

### 2.6. Diametral Tensile Strength (DTS) Test

The samples from each group (*n* = 8) were placed so that their flat ends were parallel to the base plate of the universal testing machine (Model no. 3369, Instron, Canton, MI, USA) to put the strain on the samples’ diameter (Figure 2A). A compressive force was applied using a load cell of 5 kN and a crosshead speed of 1.0 mm/min. The sample was subjected to compressive force until it fractured. The diametral tensile strength in mega Pascal was calculated using proprietary software (Bluehill ver. 2.4).

### 2.7. Compressive Strength (CS) Test

For the compressive strength testing, the samples from each study group (*n* = 8) were placed with the flat ends up between the plates of the universal testing machine (Model no. 3369 Instron, Canton, MI, USA). The sample was subjected to a compressive force with a load cell of 5 kN at a crosshead speed of 0.5 mm/min until it cracked.

### 2.8. Water Sorption and Solubility Tests

Three samples from each group were selected and stored in a desiccator with silica gel for 2 h. Next, the samples were incubated at 37 °C for 24 h, aiming to reach constant mass. The samples were weighed using a precise electronic weighing scale (Precisa, EP 320A; Dietikon, Switzerland), accurate to 0.1 mg to obtain the initial mass (m_1_) values. After achieving the initial mass, the specimens were individually immersed in a 5 mL container with distilled water and stored for 7 d. At the end of the period, samples were removed from the container and dried with blotting paper to get m_2_ values. Once again the samples were placed for dehydration in an incubator at 37 °C for 24 h and the weight measurements were repeated after 24 h (m_3_). 

The difference between the initial mass and the wet mass (m_2_ − m_1_) was used to compute the quantity of water sorption. While the difference between the initial and final drying mass values of each specimen (m_1_ − m_3_) was used to calculate the water solubility. The following formulae were used to determine the percentages of water sorption (W_sp_) and solubility (W_sol_) for each sample:W_sp_ = 100 × (m_2_ − m_1_)/m_1_
W_sol_ = 100 × (m_1_ − m_3_)/m_1_
where, V is the sample volume before immersion (mm^3^). For each group, the means and standard deviations for solubility and sorption were calculated.

### 2.9. Statistical Analysis

The acquired data were analyzed using statistical software, i.e., SPSS ver. 28 (IBM Corp., New York, NY, USA). The one-way analysis of variance (ANOVA) and Tukey’s post hoc tests were used to compare groups at a 95% confidence level (*p* < 0.05).

## 3. Results

In Figure 3, the FTIR spectrum suggests that there was a slight shift of the chemical peaks from the pure GA to the oxidized sample. It was shown that the fingerprint of GA (between 900 and 1200 cm^−1^) was changed confirming that the polysaccharide backbone was modified by the peroxide reaction. The higher absorbance at 1034 cm^−1^ could be attributed to a C-O stretches while at 1640 cm^−1^ and 1725 cm^−1^ the absorption peak could be linked to the vibration of a C=O. It could be related to the aldehyde group present. The absorption peak at 3316 cm^−1^ could be related to OH stretching of the acid groups [18].

In Figure 4, the x-ray diffraction spectra of pure GA (Figure 4A) and oxidized GA (Figure 4B) are demonstrated. Pure GA is amorphous with a few distinct peaks. While oxidized GA exhibited typical x-ray diffraction peaks particularly in the range of 10° to 50° (2 θ) indicated a high degree of crystallinity. The diffraction pattern of GA conformed with the previous published data [19].

Figure 5 displays the graphical representation of the mean and standard deviation nano hardness values of the study groups. The highest nano hardness was calculated in G2 Medicem (0.57 ± 0.16 GPa). While the lowest nano hardness was observed in G6 Medicem (0.16 ± 0.06 GPa). The two-way ANOVA model suggests materials had an insignificant effect (*p* = 0.113) on nano hardness while study groups caused a significant effect (*p* = 0.000). However, the interactive effect was observed as insignificant (*p* = 0.545).

Figure 6 displays the graphical representation of the mean and standard deviation elastic modulus values of the study groups. The highest elastic modulus was calculated in G3 Medicem (24.94 ± 5.54 GPa). While the lowest nano hardness was observed in G6 Medicem (1.53 ± 0.41 GPa). The two-way ANOVA model suggests that materials, study groups and their interactive effect all had a significant effect on elastic modulus (*p* < 0.001).

Figure 7 displays the graphical representation of the mean and standard deviation diametral tensile strength of the study groups. The highest diametral tensile strength was calculated in G2 Medicem (18.96 ± 3.08 MPa). While the lowest nano hardness was observed in G6 Ketac (7.04 ± 1.24 MPa). The two-way ANOVA model suggests that both materials and study groups had a significant effect on diametral tensile strength (*p* < 0.001). However, their interactive effect was observed as insignificant (*p* = 0.698).

Figure 8 displays the graphical representation of the mean and standard deviation compressive strength of the study groups. The highest compressive strength was calculated in G2 Ketac (60.89 ± 9.78 MPa). While the lowest nano hardness was observed in G6 Ketac (24.26 ± 2.89 MPa). The two-way ANOVA model suggests that the study groups had a significant effect on compressive strength (*p* < 0.001). However, the materials (*p* = 0.459) and the interactive effect of materials and study groups were observed as insignificant (*p* = 0.119).

The ANOVA indicated that there was a significant difference in the water solubility between the groups (*p* < 0.001). The control groups of both GICs demonstrated the lowest solubility %, i.e., 0.29 ± 0.01 and 0.13 ± 0.04 for Medicem and Ketac Cem radiopaque, respectively. However, the solubility parameters of their corresponding experimental cement with 0.5% GA formulations were also observed as insignificant compared to the control groups. Increased wt.% of GA in GICs caused to increase in the solubility %, irrespective of the GIC cement used. Similarly, water sorption of the control groups exhibited the least values compared to the experimental formulations. The details are in Table 2.

## 4. Discussion

The current laboratory study successfully synthesized different formulations of GA-incorporated GICs. The obtained data suggest enhanced physical and mechanical properties of the few experimental formulations using lower weight ratios of GA powder in GIC compared to the control group. Therefore, the hypothesis that the experimental GA-based GIC luting materials would enhance the physical-mechanical characteristics is accepted in the current investigation.

Peroxide-mediated oxidation of GA converts the polysaccharides present into various reactive oxygen species. Due to oxidation, many small and large acid groups such as glucuronic acid, galacturonic acid, glucaric acid, guluronic acid are formed (Figure 9). These acids have carboxyl (-COOH) and hydroxyl (-OH) groups attached to the glucose molecule. The hydroxyl groups of the acids bind with silica, alumina and calcium, present in surplus amounts in the powder composition of GIC. Also, the formation of acid groups lowers the pH of the oxidized GA which helps in setting the reaction and improves strength and hardness.

Because of the fluoride release, GIC is considered to exhibit cariostatic properties. The fluoride ion may prevent demineralization and bacterial growth. As a result, a luting material must impede the colonization of cariogenic bacteria at the interface of restoration and dentine for the longevity of the restoration [7,20]. Hence, GIC luting is the material of choice. Although visual examination of GIC samples incorporating different weight percentages of Gum Arabic (GA) revealed a noticeable increase in discoloration (as shown in Figure 1), the inclusion of 0.5 and 1.0 wt.% of GA in GIC exhibited a relatively minimal impact on the degree of discoloration. It is anticipated that the color change observed in 0.5 and 1.0 wt.% GA-GIC formulations would be deemed acceptable when utilized for dental restorations. In this study, we observed that modified GIC with 0.5 and 1.0 wt.% GA enhanced the nano hardness and elastic modulus. This might suggest that when GA powder is added in low weight ratios to GIC, it can enhance the nano hardness and elastic modulus of the set cement by improving the adhesion between the cement particles. Additionally, due to a high surface area, GA might conform to irregular surfaces in the set cement. Thus, improving the nano hardness. GA is a natural binding agent [21]. Due to the adhesive properties and formation of a strong interface with the glass particles, glass particles might have bonded together and enhanced the cohesion of the cement. The improved cohesion of the cement particles might help in strengthening the matrix of the GIC hence resistant to deformation under stress [22]. However, at increased weight ratios we observed detrimental effects. GA can interfere with the setting reaction of the GIC, leading to reduced conversion of the reactants and a weaker overall cement matrix. Additionally, Gum Arabic may contribute to the formation of pores or voids in the GIC matrix, which can further reduce its mechanical strength. The excess GA powder may lead to a decrease in the powder’s flowability, making it difficult to mix and handle.

The improved diametral tensile strength using low weight ratios of GA powder (i.e., 0.5, & 1.0 wt.%) in GIC powder might suggest that the addition of rubbery fillers such as GA help to reduce the brittleness of the cement by introducing a more ductile phase into the material [23]. The impregnated fillers prevent crack propagation and increase the overall strength of the material [24]. Also, the addition of GA can increase the toughness of the material by absorbing energy during deformation [25]. This results in a material that is more resistant to fracture and can withstand higher stresses and improved diametral tensile strength. While the higher wt. ratios of GA powder might lead to a reduction in the degree of conversion of the cement, which is a measure of the extent to which the liquid phase has reacted with the powder particles to form a solid network. A decrease in the degree of conversion resulted in a weaker material with lower diametral tensile strength.

Because of weak compressive strength, GIC luting has limitations for clinical use [26]. We observed that GA powder when added to GIC at low weight ratios, increased the CS. This might indicate that GA, which is hydrophilic, prevent the water escape before it became strongly bound by hydration of the cations released from the glass or siloxane groups on the surface of glass particles [27,28]. The early loss of water reduces the degree of cross-linking and increases the porosity of the cement [29], leading to a weaker CS. Additionally, GA contains water-soluble polysaccharides, which can form hydrogen bonds with the polyacrylic acid component of the cement. These hydrogen bonds help to create a stronger, more cohesive matrix within the cement, which results in a material that is more resistant to compression [30]. Also, GA contains calcium ions which can react with the glass particles in the cement to form a stronger bond. This additional bonding mechanism helps to further increase the strength of the cement and improve its overall mechanical properties.

Water sorption and solubility are critical parameters and are directly related to the longevity of cement [31]. We observed that with the increasing weight ratios of GA powder in GIC, the water solubility and sorption of the set GIC cement increased. This is because of the hydrophilic nature of the GA powder which allows it to interact with water molecules, which in turn facilitates the penetration of water into the cement matrix [32]. This increased water penetration leads to enhanced ion exchange between the glass particles and the surrounding environment, resulting in increased solubility and sorption. The presence of water can lead to a reduction in mechanical properties and deterioration of the bond between the luting agent and the tooth surface [33]. Although the G1 group (control) of both cement showed the least water solubility and sorption values. However, among the experimental groups, the G2 group of both cement showed insignificant differences compared to their corresponding G1 groups. The highest solubility and sorption % were observed in G6 groups, irrespective of the GIC used. This is because GA consists of polysaccharide that contains both arabinose and galactose sugars [34]. These sugars have hydroxyl (-OH) groups that make them highly hydrophilic and able to form hydrogen bonds with water molecules [35]. Since various techniques exist for evaluating solubility and sorption therefore it is challenging to compare our results with earlier studies.

The improved mechanical attributes of GIC using lower filler loading of GA could have several seasons such as better particle dispersion and less clustering due to which improved packing and interfacial bonding between the filler and matrix. The reaction between a rubbery powder and a weak acid may result in some level of chemical breakdown of the rubber, which could cause it to soften or dissolve to some extent leading to better particle packing. We assume that the higher filler loading of GA interferes with the hydration and setting reactions of the cement. Hence, the tested properties deteriorated.

The use of GA-reinforced GIC luting would be beneficial in patients with high caries risk or sensitivity to resin-based materials. GICs are known for their biocompatibility and low cytotoxicity. They release fluoride, which has antibacterial actions and helps prevent secondary caries and remineralize the tooth structure [4] and provides long-term protection to the tooth-restoration interface. Being easier to handle and manipulate during the cementation process and the ability to set and bond in the presence of moisture, GIC luting is an appropriate luting choice in most clinical situations.

Experimental studies are often conducted under highly controlled and artificial conditions that may not accurately reflect clinical situations. In future, the effects of different particle sizes of GA can be studied. The chemical modification of GA such as silanization or acetylation can be explored. The synergistic effects of the different fillers on the GIC matrix can be investigated.

## 5. Conclusions

The chemical characterization of the GA powder confirms the successful oxidation of polysaccharides into various reactive oxygen species. The findings of this laboratory study suggest that GA powder is a promising additive for conventional GIC luting material. The addition of GA in lower weight ratios such as 0.5 and 1.0 wt.% in GIC powder can increase the mechanical attributes of GIC luting material such as nano hardness, elastic modulus, CS and DTS. However, statistically insignificant increase in water solubility and sorption parameters using 0.5 or 1.0 wt.% GA formulations in GIC might suggest further research.

## Figures and Tables

**Figure 1 polymers-15-02679-f001:**
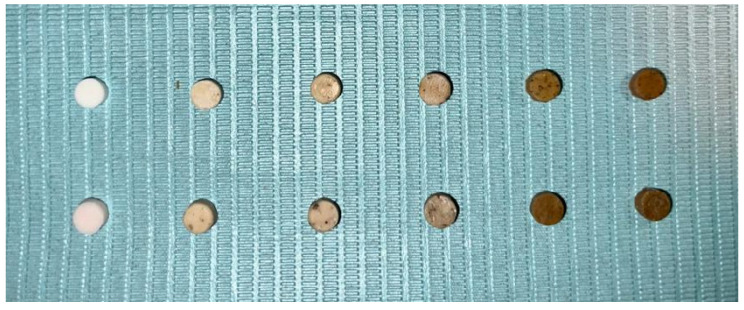
Disk-shaped study samples with 0%, 0.5%, 1.0%, 2.0%, 4.0% and 8.0% GA incorporated in GIC (from left to right). The top and bottom rows represent samples using Medicem and Ketac Cem Radiopaque, respectively.

**Figure 2 polymers-15-02679-f002:**
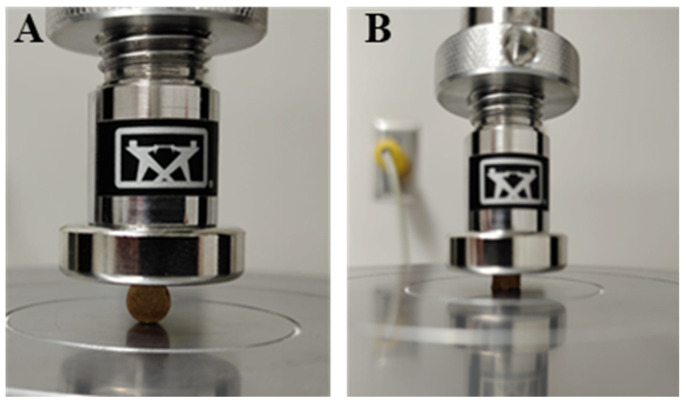
Mechanical properties evaluation of 8 wt.% GA GIC sample: (**A**) under compressive load for diametral tensile strength and (**B**) under compressive load for compressive strength.

**Figure 3 polymers-15-02679-f003:**
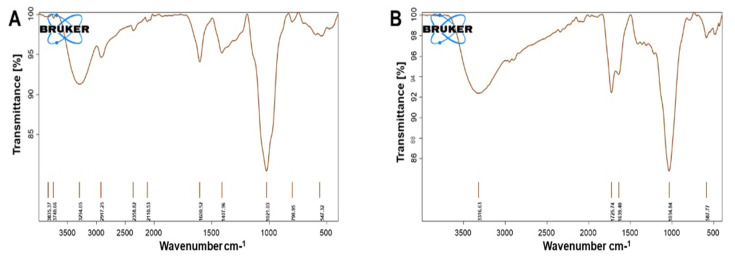
FTIR spectra: (**A**) of the unoxidized GA powder and (**B**) of the oxidized GA powder.

**Figure 4 polymers-15-02679-f004:**
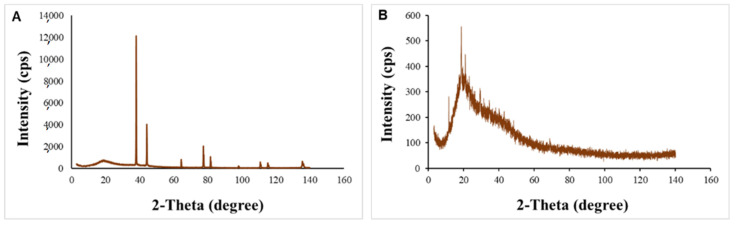
XRD patterns: (**A**) of the unoxidized GA powder and (**B**) of the oxidized GA powder.

**Figure 5 polymers-15-02679-f005:**
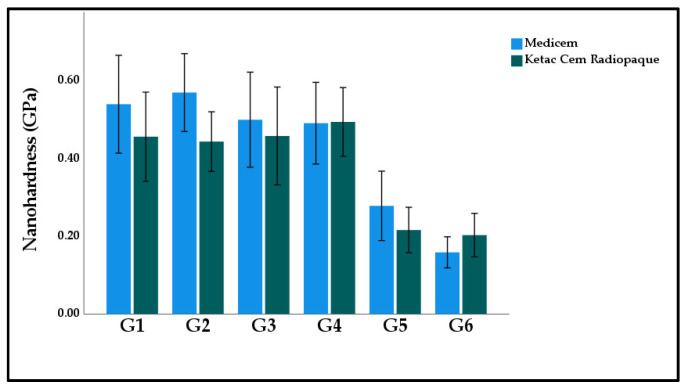
Bar graph illustrating the mean nano hardness of the experimental GICs modified with functionalized GA powder. Note: Significant differences were observed between G1:G5; G1:G6; G2:G5; G2:G6; G3:G5; G3:G6; G4:G5; G4:G6 in both Medicem and Ketac Cem Radiopaque.

**Figure 6 polymers-15-02679-f006:**
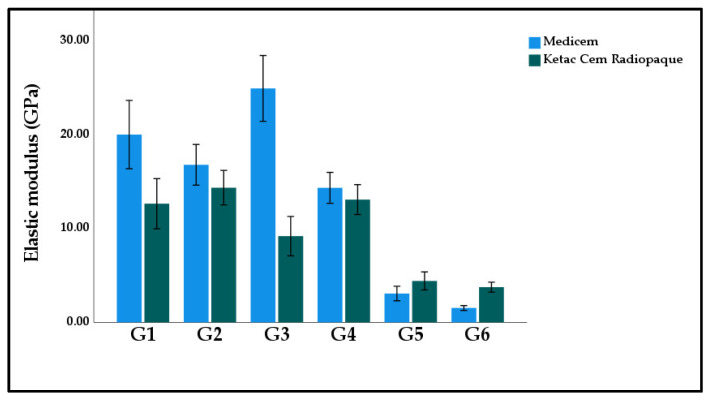
Bar graph illustrating the mean elastic modulus of the experimental GICs modified with functionalized GA powder. Note: Insignificant differences were observed between G1:G2; G2:G4; G5:G6 in Medicem while insignificant differences were observed between G1:G2; G1:G3; G1:G4; G2:G3; G2:G4; G5:G6 in Ketac Cem Radiopaque.

**Figure 7 polymers-15-02679-f007:**
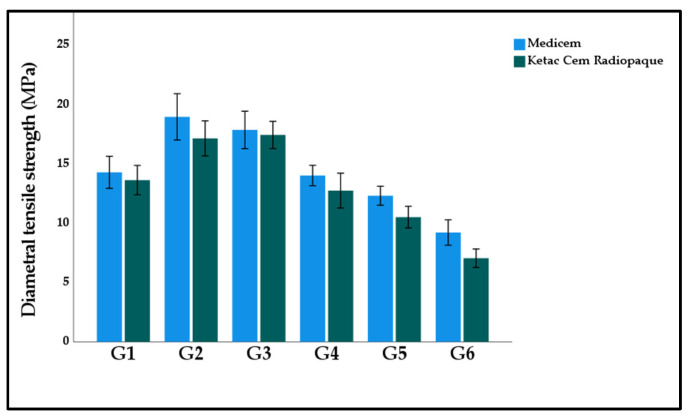
Bar graph illustrating the mean diametral tensile strength of the experimental GICs modified with functionalized GA powder. Note: Insignificant differences were observed between G1:G4; G1:G5; G2:G3; G4:G5 in Medicem while GI:G4; G2:G3; G4:G5 in Ketac Cem Radiopaque.

**Figure 8 polymers-15-02679-f008:**
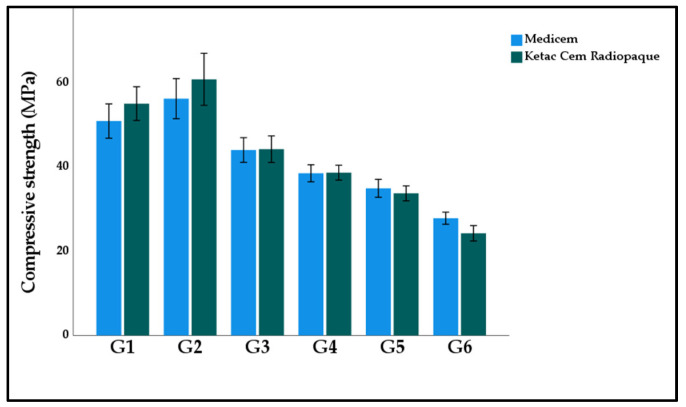
Bar graph illustrating the mean compressive strength of the experimental GICs modified with functionalized GA powder. Note: Insignificant differences were observed between G1:G2; G3:G4; and G4:G5 in both Medicem and Ketac Cem Radiopaque.

**Figure 9 polymers-15-02679-f009:**
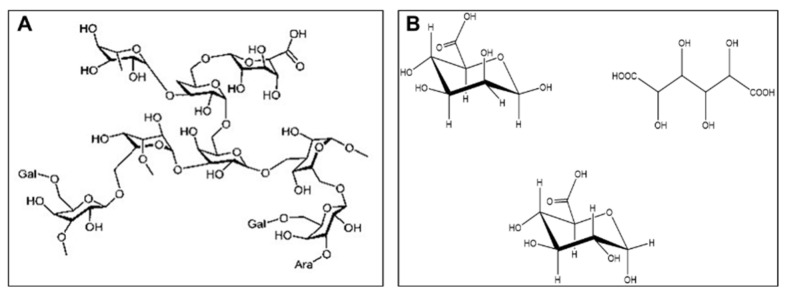
Postulated chemical structures of GA: (**A**) before oxidization and (**B**) after oxidization.

**Table 1 polymers-15-02679-t001:** The study groups with their corresponding GA wt.% and a powder:liquid ratio used for the fabrication of the samples.

Group	GIC Powder (gm)	GA wt.% in GIC	Experimental Powder Formulation (gm)	Powder:Liquid Ratio
G1 (Control)	4	0.0	4.00	1:1
G2	4	0.5	4.02	1:1
G3	4	1.0	4.04	1:1
G4	4	2.0	4.08	1:1
G5	4	4.0	4.16	1:1
G6	4	8.0	4.32	1:1

**Table 2 polymers-15-02679-t002:** Mean (±Standard Deviation) percentage change of water solubility and sorption among the control and experimental study groups.

Group	Medicem		Ketac Cem Radiopaque	
	Water Solubility (%)	Water Sorption (%)	Water Solubility (%)	Water Sorption (%)
G1 (Control)	0.29 ± 0.11 ^a^	5.82 ± 0.54 ^a^	0.23 ± 0.14 ^a^	1.69 ± 0.17
G2	0.39 ± 0.13 ^a^	6.71 ± 0.32 ^a,b^	0.36 ± 0.17 ^a^	2.18 ± 0.11 ^a^
G3	0.72 ± 0.16 ^b^	7.25 ± 0.27 ^b,c^	1.16 ± 0.24 ^b^	2.59 ± 0.20 ^a,b^
G4	0.85 ± 0.25 ^b^	7.99 ± 0.35 ^c^	1.46 ± 0.12 ^b,c,d^	2.63 ± 0.16 ^b^
G5	1.23 ± 0.10	9.23 ± 0.55 ^d^	1.68 ± 0.31 ^c,e^	3.13 ± 0.49
G6	1.51 ± 0.18	9.88 ± 0.38 ^d^	1.97 ± 0.42 ^d,e^	5.08 ± 0.40

Same lower case letters within the column depict statistically insignificant differences between the groups.

## Data Availability

The data presented in this study are available on request from the corresponding author.

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
