# Peer review of "Oxidized Natural Biopolymer for Enhanced Surface, Physical and Mechanical Properties of Glass Ionomer Luting Cement"

_polymers, 2023, doi:10.3390/polym15122679_

Round 1

Reviewer 1 Report

In this manuscript, authors investigated in improvement of mechanical properties of dental glass ionomer cement by mixture of Gum Arabic after oxidation process. Using natural biopolymer as composite filler is very acceptable for dental/medical materials. So, reviewer consider that this manuscript is acceptable to publish after miner revision as following

a)     Did authors investigate other polysaccharides instead of GA?  Please describe details the reason authors focused to GA instead of other polysaccharides.

b)    Figure 3 and 4 are very hard to see. Reviewer strongly recommend that the thickness and color of spectrum is improved in Figure 3. Also, reviewer strongly recommend that the scale of the y-axis is improved (shortened) in Figure 4.

     c) l. 250-252, What does it means of “many small and large acid groups”? In          addition, according the oxidation process in this study, reviewer consider that        “the acid” obtained is carboxy groups generally. Why does the author not describe      as “carboxylic acid”?

Author Response

REVIEWER NO. 1

Comments and Suggestions for Authors

In this manuscript, authors investigated in improvement of mechanical properties of dental glass ionomer cement by mixture of Gum Arabic after oxidation process. Using natural biopolymer as composite filler is very acceptable for dental/medical materials. So, reviewer consider that this manuscript is acceptable to publish after miner revision as following

  1. Did authors investigate other polysaccharides instead of GA? Please describe details the reason authors focused to GA instead of other polysaccharides.

Response: Thank you for your query. The reasons for using GA are mentioned in lines 65-72. In our future studies, we are planning to investigate other polysaccharides as reinforcing agents.

  1. Figure 3 and 4 are very hard to see. Reviewer strongly recommend that the thickness and color of spectrum is improved in Figure 3. Also, reviewer strongly recommend that the scale of the y-axis is improved (shortened) in Figure 4.

Response: The thickness and color of spectrum is now improved in figure 3. Additionally, the FTIR of unoxidized GA has also been added to differentiate between the unoxidized and oxidized spectra. The scale of the y-axis is also improved in Figure 4 now. Also, the additional XRD pattern of pure GA is included for comparison.

  1. c) line 250-252, What does it means of “many small and large acid groups”? In addition, according the oxidation process in this study, reviewer consider that “the acid” obtained is carboxy groups generally. Why does the author not describe as “carboxylic acid”?

Response: Thank you for highlighting this silly mistake. We have now mentioned the names of a few acid groups formed after oxidization, also we have mentioned about the formation of carboxylic acid.

Reviewer 2 Report

This study investigates the impact of oxidised gum arabic (GA) on selected physical and mechanical properties of two commercial glass-ionomer luting cements. The study is limited, as the authors have only provided data at 24 h, by which time the maturation reactions of the GICs are still continuing and, accordingly, the mechanical properties are still developing. In general, the information is clearly described in the manuscript, although minor grammatical and formatting errors need to be addressed at the copy editing stage.

The authors should provide a rationale for oxidising the GA prior to its incorporation into the GICs. They should also include images of the ideal chemical structures for the original and oxidised GA, as many readers interested in dental materials will not have knowledge of carbohydrate chemistry. Additionally, the authors should provide FTIR and XRD data of the original unoxidized GA for comparison. SEM images of both original and oxidised GA should also be included in the manuscript, as the dimensions of reinforcing fibres influence the mechanical properties of composites.

The images in Fig. 1 should be presented in the Results, rather than the Materials and Methods, and the authors must comment on the clinical relevance of the discolouration that occurred on incorporation of GA.

The exact quantity of iron sulphate must be given in Line 87.

In Section 2.4, the authors must specify whether GA additions were calculated on the basis of the mass of powder or the total mass of powder and solution in the GIC formulation. The average masses of GA, GIC powder and GIC solution should be given in Table 1.

The authors must provide references for the FTIR assignments in Fig. 3. They should also assign all of the vibrations in the spectrum (not just three selected signals).

The authors must replot the XRD data so that the reflections are clearly legible. They must also give details of the methods used to determine crystallinity and crystallite size in Section 2.3. The statement, ‘Defined coordinates and the straight baseline confirm that the powder is crystalline.’ must be removed and replaced with an appropriate analysis of the XRD data (including any reference diffraction files used to confirm the diffraction pattern of the GA sample). The degree of crystallinity and mean crystallite size of the GA should be given. Ideally, the XRD data should be indexed, but I don’t insist on this.

The error bars in the figures do not correspond with the errors stated in the text. Furthermore, the experimental data are presented such that there appears to be no significant increase in any of the mechanical properties of the GICs on addition of any quantity of GA. The authors must address this problem.

The statistical analysis is unconvincing. For clarity, the authors should tabulate the statistical data in an appendix.

I have no confidential comments for the editors. 

In general, the information is clearly described in the manuscript, although minor grammatical and formatting errors need to be addressed at the copy editing stage.

Author Response

REVIEWER NO. 2

Comments and Suggestions for Authors

This study investigates the impact of oxidised gum arabic (GA) on selected physical and mechanical properties of two commercial glass-ionomer luting cements. The study is limited, as the authors have only provided data at 24 h, by which time the maturation reactions of the GICs are still continuing and, accordingly, the mechanical properties are still developing. In general, the information is clearly described in the manuscript, although minor grammatical and formatting errors need to be addressed at the copy editing stage.

Response: Thank you

 The authors should provide a rationale for oxidising the GA prior to its incorporation into the GICs. They should also include images of the ideal chemical structures for the original and oxidised GA, as many readers interested in dental materials will not have knowledge of carbohydrate chemistry. Additionally, the authors should provide FTIR and XRD data of the original unoxidized GA for comparison. SEM images of both original and oxidised GA should also be included in the manuscript, as the dimensions of reinforcing fibres influence the mechanical properties of composites.

Response: The rationale for oxidizing GA prior to its use in GIC is now further elaborated in the introduction section. The images of the ideal chemical structures for pure GA and oxidized GA are also incorporated in the manuscript (as Fig. 9). The FTIR and XRD data of the unoxidized GA is now also included. However, due to unavailability of SEM device, we were unable to get images of the GA powder at high magnification.

The images in Fig. 1 should be presented in the Results, rather than the Materials and Methods, and the authors must comment on the clinical relevance of the discolouration that occurred on incorporation of GA.

Response: Thank you. Yes, we have now discussed the clinical relevance of Fig. 1 in the Discussion part.

The exact quantity of iron sulphate must be given in Line 87.

Response: We have now mentioned the catalytic quantity of iron sulphate in line 88.

 In Section 2.4, the authors must specify whether GA additions were calculated on the basis of the mass of powder or the total mass of powder and solution in the GIC formulation. The average masses of GA, GIC powder and GIC solution should be given in Table 1.

Response: We have already mentioned that the addition of GA was calculated on the basis of the mass of GIC powder in the text. For the ease of readers, now we have modified Table 1 and included the weight of GIC powder, GA and the weight of GA-GIC experimental formulation.

The authors must provide references for the FTIR assignments in Fig. 3. They should also assign all of the vibrations in the spectrum (not just three selected signals).

Response: We have now cited the reference as well as defined all the vibrations in the spectrum.

The authors must replot the XRD data so that the reflections are clearly legible. They must also give details of the methods used to determine crystallinity and crystallite size in Section 2.3. The statement, ‘Defined coordinates and the straight baseline confirm that the powder is crystalline.’ must be removed and replaced with an appropriate analysis of the XRD data (including any reference diffraction files used to confirm the. The degree of crystallinity and mean crystallite size of the GA should be given. Ideally, the XRD data should be indexed, but I don’t insist on this.

Response: We have replotted the XRD data of both pure and oxidized GA powder using the raw data. The paper has been cited for the methodology used to determine crystallinity and crystallite in section 2.3. The statement “defined coordinates and the straight baseline…………” has been removed. The XRD diffraction patterns have now been conformed with the previous published data.

The error bars in the figures do not correspond with the errors stated in the text. Furthermore, the experimental data are presented such that there appears to be no significant increase in any of the mechanical properties of the GICs on addition of any quantity of GA. The authors must address this problem.

Response: We apologise for this confusion. We have now replotted the graphs (from Fig.5-8) using “standard error”. The previous plotted graphs were with “standard deviation” which is why it was showing high.

The statistical analysis is unconvincing. For clarity, the authors should tabulate the statistical data in an appendix.

Response: After using “standard error” instead of “standard deviation” in the graphs, the graphs are corroborating with the text.

I have no confidential comments for the editors.

Response: Thank you

Comments on the Quality of English Language

In general, the information is clearly described in the manuscript, although minor grammatical and formatting errors need to be addressed at the copy editing stage.

Response: Thank you

Round 2

Reviewer 2 Report

The authors have adequately addressed the major concerns, and accordingly, I recommend that the paper can be published (subject to minor grammatical and formatting refinements by the copy editor).

Minor grammatical and formatting refinements are required at the copy editing stage.